# Research on Classification of Tibetan Medical Syndrome in Chronic Atrophic Gastritis

**Xiaolan Zhu [1], Lei Zhang [2,*], Yuan Zhang [1,*], Lu Wang [1], Shiying Wang [1] and Ping Liu [1]**

[1]    Department of Computer Technology and Applications, Qinghai University, Xining 810016, China;
       zxlanscu@126.com (X.Z.); wlqgl@126.com (L.W.); wangsy706@sina.com (S.W.);liuping131327@163.com (P.L.)
[2]    College of Computer Science, Information Management Center, Sichuan University, Chengdu 610065, China
[*]    Correspondence: zhanglei@scu.edu.cn (L.Z.); 2011990029@qhu.edu.cn (Y.Z.); Tel.: +86-139-8188-4494 (L.Z.)

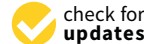

**Featured Application: In this paper, we proposed a classification model of Tibetan medical syndrome based on atomic classification association rules to provide effective decision-making support for the diagnosis and treatment of common plateau diseases more scientifically.**

**Abstract:** Classification association rules that integrate association rules with classification are playing an important role in data mining. However, the time cost on constructing the classification model, and predicting new instances, will be long, due to the large number of rules generated during the mining of association rules, which also will result in the large system consumption. Therefore, this paper proposed a classification model based on atomic classification association rules, and applied it to construct the classification model of a Tibetan medical syndrome for the common plateau disease called Chronic Atrophic Gastritis. Firstly, introduce the idea of "relative support", and use the constraint-based Apriori algorithm to mine the strong atomic classification association rules between symptoms and syndrome, and the knowledge base of Tibetan medical clinics will be constructed. Secondly, build the classification model of the Tibetan medical syndrome after pruning and prioritizing rules, and the idea of "partial classification" and "first easy to post difficult" strategy are introduced to realize the prediction of this Tibetan medical syndrome. Finally, validate the effectiveness of the classification model, and compare with the CBA algorithm and four traditional classification algorithms. The experimental results showed that the proposed method can realize the construction and classification of the classification model of the Tibetan medical syndrome in a shorter time, with fewer but more understandable rules, while ensuring a higher accuracy with 92.8%.

**Keywords:** Tibetan medical syndrome; atomic classification association rules; relative support; partial classification; classification and prediction

## 1. Introduction

Tibetan medicine belongs with a long history to traditional national medicine. As an important part of Chinese medicine, it has a unique theoretical system and clinical efficacy, which is especially effective for the treatment of digestive diseases. As a result, Tibetan medicine becomes the first choice to seek medical advice for the farmers and herdsmen in northwestern China. As a typical gastrointestinal disease in Tibetan medicine characteristics on the plateau, Chronic Atrophic Gastritis (CAG) has a high incidence and repeated lingering condition, and the risk of cancer increases when it is associated with intestinal metaplasia and dysplasia, which leads to a difficulty in clinical treatment [1]. Regarded as one of the precancerous lesions of Gastric cancer [2], CAG has attracted great attention in Tibetan medicine.

In traditional Chinese medicine syndrome differentiation, it is the comprehensive analysis of clinical information gained by the four main diagnostic procedures: Observation, listening, questioning,

and pulse analysis, which will be used for guiding the choice of treatment with Fufang, and for further stratification of the patients' conditions with certain diseases [3].

However, because of the complexity of the regional environment and cultural background, the development of Tibetan medicine has a late start and a low starting point, which is incompatible with the multi-level and diversified medical care need of the masses of people. Under new situations, it still faces many new problems and difficulties. Firstly, the lack of a unified standard library of Tibetan medicine, while the terminology and diagnosis process of Tibetan medicine results in difficulties in communication and low degree of credibility in scientific research. Secondly, the clinical research foundation is weak, and the clinical treatment technology is not standardized, which makes the original data set incomplete. Finally, there is no clinical guiding principle based on Tibetan medicine theory, and the strong doctors' subjective factors existing during the process of disease diagnosis and treatment has restricted the diagnosis and treatment of this disease to a large extent.

Recently, data mining technology has been widely used in traditional Chinese medicine, which has also promoted the development of Tibetan medicine. Since there are many differences between Tibetan medicine and traditional Chinese medicine during the process of the clinical diagnosis and treatment of disease, some data mining methods suitable for traditional Chinese medicine cannot be directly applied to Tibetan medicine. As a result, the research on data mining in the field of Tibetan medicine is still in its infancy.

Therefore, this paper put forward a classification model innovatively based on the atomic classification association rules in view of the clinical diagnosis and treatment data of common plateau disease with CAG. The classification model of the Tibetan medical syndrome is established after mining the strong atomic classification association rules implied between symptoms and syndrome, so as to provide effective decision-making support for the diagnosis and treatment of common plateau diseases more scientifically, and achieve the evidence-based intelligence of the diagnosis and treatment by Tibetan medicine.

## 2. Related Works

### 2.1. Association Rules

Association rules aim to find the implicit and interesting rules hidden in the data set through the analysis of massive data. The most famous association rules mining algorithm is the Apriori algorithm [4], that was first proposed by R. Agrawal, who came from the IBM Almaden Research Center. Later, many scholars improved the process of mining the frequent item-sets which would cost a long time usually, and some variants of the Apriori algorithm [5–7] were proposed.

Association rules algorithms have been widely used in traditional Chinese medicine. Dogan et al. [8] used data mining to find association rules for discovering hyperlipidemia from the biochemistry blood parameters, which will be very helpful for physicians in the diagnosis of this hyperlipidemia. A cross-sectional study to explore the association among the presence of immune reconstitution and use of stavudine, didanosine and protease inhibitors with thyroid diseases was conducted in the paper [9]. Association rules and decision tree algorithms were used in the tumor data set to support medical decisions in early detection of tumors, and applied to 7544 medical diagnosis cases from more than 10 medical centers in Basra, Iraq [10]. The support and confidence criteria in association rules is applied to quantify the association among semantic features in the pulmonary nodule database (the Lung Image Database Consortium—LIDC), which improved the ability for clinical radiologists to diagnose malignant pulmonary nodules after introducing new assessment indicators that related to the individual characteristics [11]. Chan et al. [12] used association rules to find disease patterns of Metabolic syndrome-related disease, and found that patients with Metabolic syndrome have higher connection with liver diseases than patients with Diabetes Mellitus.

Amato et al. [13] put forward a novel OSN data model that supports easy management of multimedia content in a unique framework, and realized a more effective and efficient mechanism

for data and information management, which will provide a way of data management during the construction of a knowledge base by using Association rules. Yi [14] pointed out the close cooperation model between traditional Chinese medicine and data-mining experts, which can solve the clinical and theoretical difficulties in traditional Chinese medicine.

In the field of Tibetan medicine, association rules are mainly used to explore the relationship between the compatibility of prescriptions and medications in Tibetan medicine, find out the regular patterns and theoretical basis of medication, and provide evidence for clinical diagnosis and treatment of Tibetan medicine. Luosang et al. [15] used association rules to analyze the frequency of drugs, implied rules, core combination and new formulae, which could provide scientific evidence for further standardization of the Tibetan medicine of gastric disease. Knowledge discovery was applied to find the medication law of Tibetan medicine treatment of plateau disease, and reveal the mechanism of plateau disease [16]. Cairang et al. [17] used association rules to explore the prescriptions for the treatment of heat-induced liver disease based on the prescriptions that come from four medical books, and proved that the theory of nature, efficacy and taste of Tibetan medicine have a strong guiding significance for clinical drug use, individualized treatment, and the research and development of new drugs. The frequency of drug use, drug combinations with high frequency, and sequence association rules hidden in medical records, were explored after analyzing drug use from the medical records of patients with stagnation disease (Cerebral Infarction) treated by the Tibetan Hospital of Tibet Autonomous Region [18]. Yang et al. [19] adopted the Apriori algorithm to summarize the regular drug use, characteristics of scorpion and clinical prescriptions of Hankezi after the analysis of common drugs, frequency of drug combinations and core drug combinations of Hankezi in Tibetan medicine, and provided support for its clinical applications.

## 2.2. Classification Analysis

Classification Analysis aims to map the instance in a data set to a given category. Classification algorithms are widely used in medical data analysis of traditional Chinese medicine. Xu et al. [20] used a decision tree algorithm to classify the biological arousal signals, by taking the demographic information of the patients, the biofeedback information of the emotional images, and the physiological characteristics of the autonomic nerves as input characteristics. Deep learning was used to construct the syndrome diagnostic model for chronic gastritis (CG) in TCM, and the results showed that deep learning could improve the accuracy of syndrome recognition [21]. Kavakiotis et al. [22] conducted a systematic review of the applications of data mining techniques and tools in the field of diabetes research, with respect to prediction and diagnosis of diabetic complications, and found that 85% of correlated articles used supervised learning approaches, and association rules, where the Support Vector Machine arises as the most successful and widely-used algorithm. Aswal et al. [23] performed experimental analysis of traditional classification algorithms such as Support Vector Machine on bio-medical datasets to validate their performance. K-means algorithm was used to find significant groups of patients, and Apriori algorithm was applied to understand the kinds of relations between the attributes, and Multilayer Perceptron Neural Network was adopted to predict CVD risk [24]. A lexicon-grammar based methodology for efficient information extraction and retrieval on unstructured medical records, and describing the NLP methodology for extracting RDF triples from these records, was proposed [25]. Application of deep learning for medical diagnosis was addressed, and found that convolutional neural networks (CNN) are the most widely represented in medical image analysis in the paper [26].

In research of Tibetan medicine, classification algorithms can be used for the correlation analysis and prediction between diseases and symptoms. Li et al. [27] reviewed the application of data mining in the field of Tibetan medicine, particularly for clustering, association rules and classification technology. Li et al. [28] applied the artificial neural network to the clinical research of Tibetan medicine for the treatment of Hematorrhea, and the association relationship between Hematorrhea and hypertension was discovered.

Research of the compatibility of Tibetan medicine prescriptions with the use of association rules and classification methods was proposed, which used drug factors combined with the nature and location of the disease [29]. Wang et al. [30] proposed a gray box method based on a distance-K-nearest neighbor algorithm that combined with the nature and location of the disease, by using clustering and association analysis of CAG, and a predictive model with an accuracy of 80.1% for the diagnosis and treatment of CAG was established after considering a patient's individual and symptom characteristics.

### 2.3. Association Classification

Association classification algorithm is a new classification method that integrates association rules with classification. It predicts the class of unknown instance by mining the strong association rules that meet the minimum support and confidence thresholds. Compared with traditional classification algorithms, it has higher classification accuracy and stronger adaptability. Liu et al. [31] first proposed the association classification algorithm called CBA, which uses an Apriori algorithm to generate classification association rules by setting the minimum support to 0.1, and the minimum confidence to 0.5. However, it cost a lot of time to obtain the candidate frequent item-sets, since that Apriori algorithm needs to scan the database many times. Li et al. [32] put forward a classification algorithm called CMAR, which was based on multiple classification association rules, and determined the class of unknown object in view of multiple classification association rules that came with different weights. Yin [33] proposed a predictive classification algorithm called CPAR, which generated fewer rules than CMAR, but an equivalent accuracy. Vo and Le [34] found all classification association rules and pruned redundant rules to gain the smaller rules set based on an ECR-tree. A multi-minimum support model was applied to provide user-definable minimum support for the rules items in the database, so as to mine rare items effectively [35]. Li et al. [36] analyzed medical and healthcare data comprehensively from both positive and negative association rules based on the medical data collected from a person's Hospital, which can provide an important reference value for medical research. Association algorithm with the atomic combination of drugs was applied to explore drugs rules based on the pulse, urine, and tongue diagnosis data of chronic atrophic gastritis to make up for the deficiencies in the research of the prescriptions of Tibetan medicines [37].

However, the above algorithms based on association rules for classification have the following problems, such as the large number of generated rules, the slow speed of execution, the long time for constructing the classification model and the incomprehensible classification rules, etc. Therefore, from the perspective of data mining, this paper proposed a classification model based on atomic classification association rules, which mines the strong atomic classification association rules implied between symptoms and syndrome, and tries to achieve the rapid classification and prediction of Tibetan medical syndrome with less and more understandable rules, while maintaining a higher accuracy, and further provides supportive decision support for Tibetan medical practitioners to diagnose and treat common diseases of plateau.

## 3. Atomic Class Association Rules

The atomic class association rules (ACAR) are some implication rules that like x => y, where x is the predecessor, and y is the posterior of the rule. In addition, both the predecessor and the posterior have only one attribute, and the posterior is fixed as a classification attribute. It is a subset of association rules, and can reflect the intrinsic relevance and the predictability of rules, and can be used to distinguish the class of data objects in the data set. Its goal is to mine the constrained atomic classification association rules that satisfy the support and confidence threshold.

The following is a formal definition of some related concepts mentioned in this paper during the process of the mining of atomic classification association rules.

At first, suppose there is an initial data set called D, I is a collection of all items in D, and C is a collection of all of the class labels. Besides, there is a data set of a partial classification that called $D_i$.

Then, there will be the following definition:

**Definition 1.** *ACAR, The form of the rule r is: x => y, where |r.x| = |r.y| = 1, x ∈ I, y ∈ C.*

**Definition 2.** *The support of ACAR. The ratio of the number of instances meet r whose predecessor is x and classification attribute is y in D to the total number of all instances in D, and the formula is as Equation (1):*

$$Sup(r) = \frac{count(r.x \cap r.y)}{|D|} \tag{1}$$

**Definition 3.** *The confidence of ACAR. The ratio of the number of instances satisfies r whose predecessor is x and classification attribute is y in D to the total number of instances that take x as the predecessor in D, and the formula is as Equation (2):*

$$Conf(r) = \frac{count(r.x \cap r.y)}{count(r.x)} \tag{2}$$

**Definition 4.** *Relative support of ACAR. The ratio of the number of instances meet r whose predecessor is x and classification attribute is y in $D_i$ to the total number of instances in $D_i$ during the process of the i-th partial classification, and the formula is as Equation (3):*

$$SupR(r) = \frac{count(r.x \cap r.y)}{|D_i|} \tag{3}$$

**Definition 5.** *Strong ACAR.*

$$|r.x| = |r.y| = 1, \ x \in I, \ y \in C$$

$$SupR(r) >= minSup$$

$$Conf(r) >= minConf$$

*The strong ACAR are the rules that satisfy the above three conditions, and the minSup and minConf represent the minimum support and confidence threshold specified by users in advance, respectively.*

## 4. Constructing Classification Model of Tibetan Medical Syndrome

After the analysis of Tibetan medicine classics and clinical diagnosis and treatment data, we can find that there is a close and complex relationship between symptoms and syndrome in disease that the clinical and checked characteristics reflected through symptoms are related to the syndrome of disease (see Figure 1).

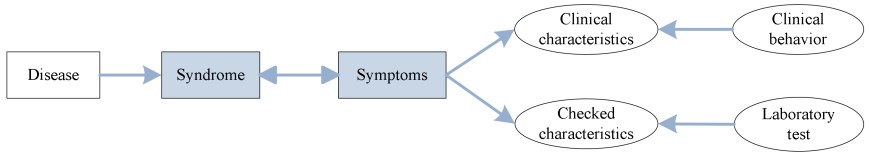

**Figure 1.** Relationship between symptoms and syndrome.

Therefore, the flow chart of the classification model of Tibetan medical syndrome based on ACAR is as shown in Figure 2. Firstly, data preprocessing technology is adopted to obtain a standardized data set suitable for later data mining after analyzing the classics and clinical diagnosis data of Tibetan medicine. Secondly, a constraint-based association rules mining algorithm that combines the idea of "relative support" is applied to mine the implicated strong atomic classification association medical diagnosis rules between symptoms and syndrome, so as to build the knowledge base of Tibetan medical diagnosis and treatment. Finally, the classification model is established after pruning and prioritizing

these rules. At the same time, after introducing the idea of "partial classification", the prediction of the syndrome of Tibetan medicine is achieved based on "significant characteristics" and "the first easy and difficult" strategy in order to provide effective decision support for doctors to diagnose and treat this Tibetan medical syndrome more accurately.

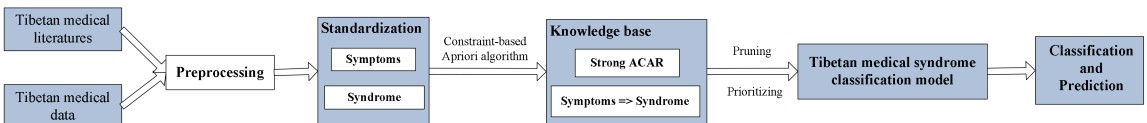

**Figure 2.** Flow diagram of classification model of the Tibetan medical syndrome.

### 4.1. Preprocessing

Since there are both continuous and discrete attributes in the original data set of Tibetan medical syndrome, and the data set lacks some symptoms, and there are also noises and inconsistency etc., the data set needs to be preprocessed for further analysis. First of all, the records with more than 80% of missing attributes are deleted, and the continuous and discrete symptoms with missing data are respectively filled with the mean and mode of the symptoms. Simultaneously, the continuous symptoms are discretized, and the noises are smoothed in order to obtain the standardized data set suitable for the data mining in this paper.

Furthermore, the filtered feature selection algorithm (CFS) is used to improve the precision of classifier, and it realizes the feature selection and dimension-reduction based on the importance of the feature to classifier, and the correlation with other features, and the formula is as Equation (4).

$$F_s = \frac{n\overline{rca}}{\sqrt{n + n(n-1)\overline{raa}}} \tag{4}$$

where, s is the feature set, which consists of n attributes, $\overline{rca}$ is the average correlation between the attribute a and the classifier c, and $\overline{raa}$ is the average correlation among the attributes in the feature set. The numerator represents the predictive ability of the feature set to the classifier, and the denominator represents the redundancy between the attributes in the feature set. Finally, $F_s$ is the classification capability of the classifier after removing redundant features from the feature set.

### 4.2. Mining Strong ACAR

Taking the symptoms in the data set as the conditional attribute and the Tibetan medical syndrome as the decision attribute, and further using the constraint-based Apriori algorithm to mine the strong ACAR like "symptoms => syndrome", as is shown in Algorithm 1.

---

**Algorithm 1:** Mining strong ACAR.

---

**Input**: Suppose there is a partial classification data set called $D_i$, an attribute set called A, and a class label set called C.
**Output**: A strong ACAR set called rulesSet.

---

- CandidateRulesSet = ∅
- RulesSet = ∅
- FOR k=1 to |C| DO
- 　FOR I=1 to |A| DO
- 　　FOR j=1 to |dom($A_i$) | DO
- 　　　R = {$A_i$ = value($A_i$,j) => C = value(C,k)}
- 　　　SupR(r) = count(r.xˆr.y) / |$D_i$|
- 　　　Conf(r) = count(r.xˆr.y) / |count(r.x)|
- 　　　IF SupR (r) >= minSup THEN
- 　　　　candidateRulesSet = candidateRulesSet ∪ {r}
- 　　END FOR
- 　END FOR
- END FOR
- FOR r in | candidateRulesSet |
- 　IF Conf(r) >= minConf THEN
- 　　rulesSet = rulesSet ∪ {r}
- 　END IF
- END FOR

---

Among them, minSup and minConf represent the minimum support and confidence threshold, respectively. $A_i$ represents the i-th attribute and dom () is the domain of the attribute, value ($A_i$, j) represents the j-th value belonging to the attribute $A_i$, and count () is used for counting.

### 4.3. Constracting the Classification Model of Tibetan Medical Syndrome

Since the rules in the strong ACAR set are redundant and have priority, so these redundant rules need to be pruned in order to improve the accuracy of the classification model, and those ACAR with high priority will be selected to establish the classification model.

Firstly, prune the rules in the strong ACAR set based on the ability to correctly classify samples. For each rule r in the strong ACAR set, it is detected whether there are instances in the training set covered by r. If r can correctly classify at least one instance, then r will be marked as a potentially useful atomic classification association rule. When the rules in the strong ACAR set or the instances in the training set are all traversed, the rule that cannot correctly classify any instance is cut off, and the pruning process of the rule will be ended. Secondly, prioritize the ACAR by using the confidence as the first keyword, and the support as the second keyword (Conf1Sup2), correlation coefficient (Corr), and lift (Lift), respectively, and a set of ACAR with high priority will be selected to establish the classification model. Finally, build the syndrome classification model that consisted of the selected potential useful ACAR based on the priority of rules and default_class.

### 4.4. Predicting Tibetan Medical Syndrome

The key of classification techniques based on ACAR is atomicity, which mimics the human classification of things based on the implicit association between the distinctive features of the object and the category attributes. However, it is difficult to classify all instances, once they have been based on a few significant features.

Therefore, it is necessary to introduce the idea of "partial classification", and use the strategy of "easy first and then difficult" to perform multiple classifications until all instances in the data set are classified.

At first, all ACAR are mined for the classification of the test data set based on the initial data set. Then, mine the ACAR in the remaining unclassified instances based on the relative support continuously, to make the original features that are not significant become salient features in order to realize the partial classification, and the partial classification process is repeated until all instances in the test data set are classified. At last, calculate the number of instances that are correctly classified by the classification to obtain the accuracy of the classification model. Furthermore, the classification model with high precision is selected for doctors to diagnose the syndrome of disease more accurately and intelligently during the process of the clinical diagnosis and treatment of Tibetan medicine, thus improving the level of the clinical diagnosis and treatment of common plateau disease.

## 5. Experiment and Analysis

### 5.1. Data Set

The experimental data in this paper is derived from the Tibetan medical clinical data of 218 patients with CAG provided by the National Natural Science Foundation project cooperation unit (Tibetan Hospital of Qinghai Province). The original data is converted into a standard data set consisting of 208 complete medical records which are suitable for data mining after data preprocessing. Finally, the symptoms mainly contain gastroscopy, pathological biopsy, blood, pulse, urine, tongue diagnosis and general symptoms, etc. Totally, there are 76 attributes.

### 5.2. Results of the Proposed Method

Firstly, take syndrome as the classification attribute, the key symptom factors are selected with the CFS feature selection algorithm, and the constraint-based Apriori algorithm is used to mine the strong ACAR of "symptoms => syndrome" with minSup 0.05 and minConf 0.8 (see Figure 3). Where, Syndrome 1, Syndrome 2, Syndrome 3, and Syndrome 4 represent the four kinds of syndromes, and the size and color of the circle represent the support and lift of each rule, respectively.

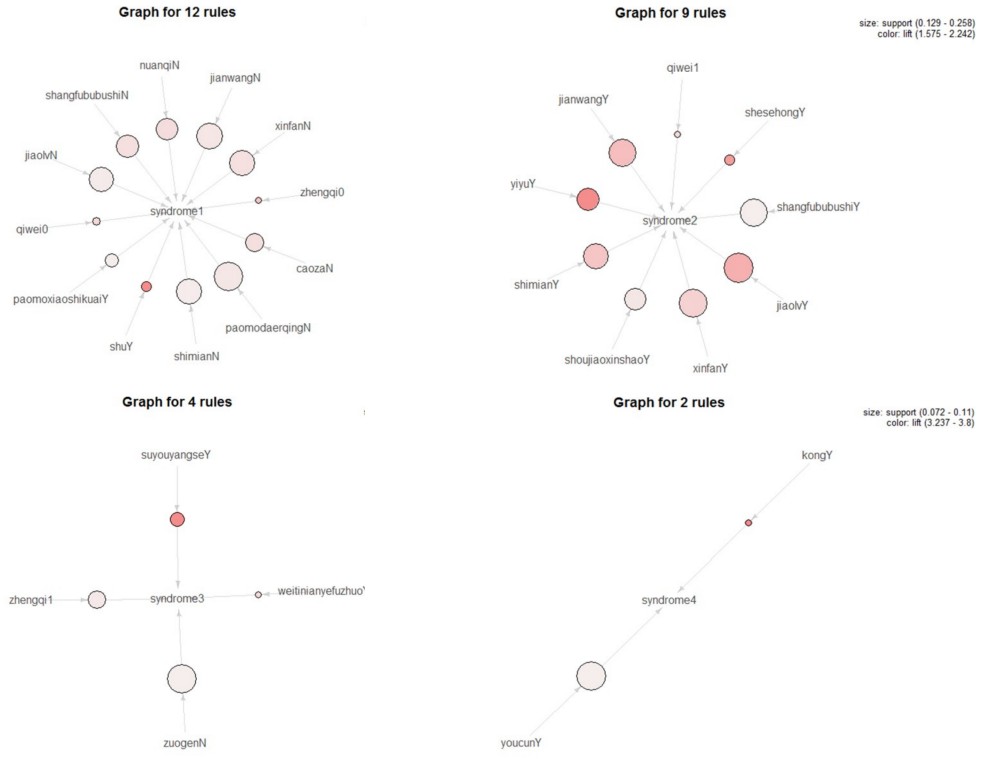

**Figure 3.** Visualization of the strong ACAR.

Secondly, sort the priority of the ACAR based on Conf1Sup2, Corr and Lift sorting methods, respectively. Then, the corresponding classification models are established. Finally, based on the above three rule sorting methods, the times of partial classification and the effectiveness of the classification model are verified after taking all data sets (All), according to the 3:2 random selection (Random), sequential selection (Order), five-fold cross-validation (5-Folds-Validation), which are the four ways to construct the training set and test set. The classification accuracy is as shown in Table 1 and Figure 4.

**Table 1.** Partial classification results of the proposed method.

| Rules Sorting Methods; Data Set Building Ways | Accuracy (%) | | | | | |
| --- | --- | --- | --- | --- | --- | --- |
| | First Partial Classification | | | Second Partial Classification | | |
| | Conf1Sup2 | Corr | Lift | Conf1Sup2 | Corr | Lift |
| All | 87.5 | 89.4 | 90.4 | 87.5 | 89.4 | 90.4 |
| Random | 88.4 | 91.3 | 91.3 | 88.4 | 91.3 | 91.3 |
| Order | 88.4 | 89.9 | 92.8 | 88.4 | 89.9 | 92.8 |
| 5-Folds-Validation | 86.5 | 89.4 | 90.4 | 87.5 | 89.4 | 90.9 |

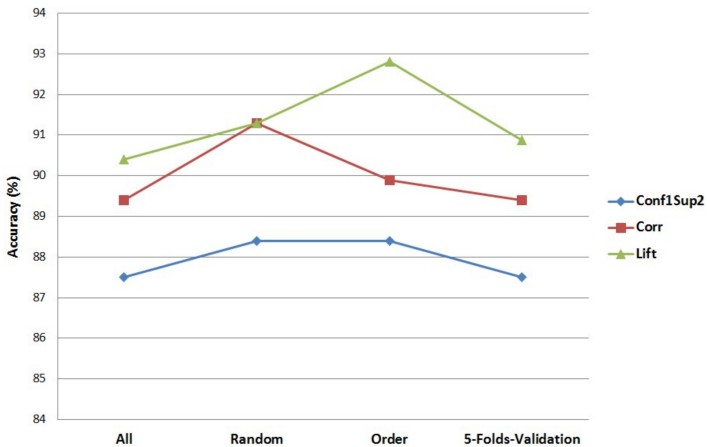

**Figure 4.** Classification accuracy of the proposed method.

It can be seen that the proposed method can realize the classification of all instances in the data set with only two times of partial classification, and the first partial classification can classify most samples. In addition, no matter which training set and test set construction method is adopted, the classification model based on the Lift sorting method has the highest accuracy, and the accuracy of the training set and test set constructed based on Order, can be up to 92.8%.

*5.3. Comparison with CBA Algorithm*

In order to compare with the CBA algorithm, the minimum support minSup is set to be 0.05 and the minimum confidence minConf is set to be 0.8, and with the same experimental environment. Firstly, construct the training set and the test set with All, Random, Order, and 5-Folds-Validation four ways, and the ACAR are mined. Then, sort the priority of the rules based on the Lift optimal sorting method, and 40, 80, 200, 400, 800, 1000 rule sets of different sizes are selected according to the priority order to establish the corresponding classification models. Finally, validate the accuracy of the classification model with the CBA algorithm, as is shown in Figure 5.

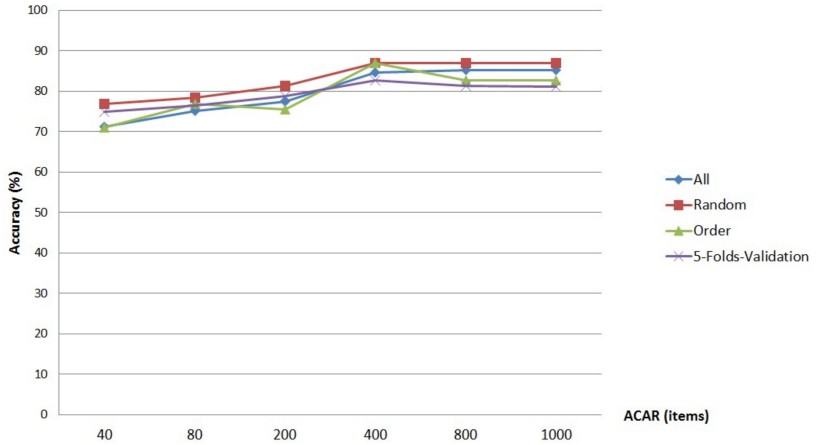

**Figure 5.** Classification accuracy based on CBA algorithm.

It can be seen that the accuracy of the classification model is the highest when the number of the ACAR is 400, and the training set and test set are constructed based on the Random method. Under this condition, the performance of classification model is compared with the proposed method from the following three aspects: The number of atomic classification association rules generated, the time consumption to build the classification model and predict new instances and accuracy, and the result as is shown in Table 2.

**Table 2.** Comparison of classification performance between the proposed method and CBA.

|                    | ACAR (Items) | Time Consumption (ms) | Accuracy (%) |
| ------------------ | ------------ | --------------------- | ------------ |
| CBA algorithm      | 400          | 955                   | 87           |
| The proposed method| 27           | 220                   | 92.8         |

It is obvious that the classification model based on the proposed method can classify all samples in the data set in a shorter time, which requires only 27 strong atomic class association rules, and two times of partial classification, and the accuracy is 92.8%, which is higher than the CBA algorithm.

*5.4. Comparison with Traditional Classification Algorithms*

In order to verify the effectiveness of the proposed method, the accuracy of the four traditional classification algorithms (KNN, J48, Bayes, Random Forest) is compared, based on the five-fold cross-validation method (see Figure 6).

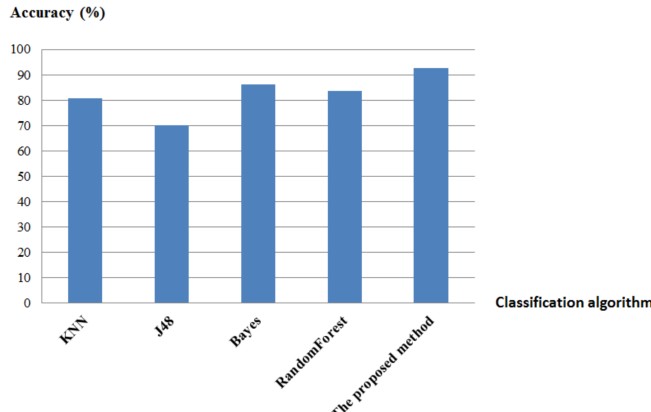

**Figure 6.** Comparison of accuracy between the proposed method and four traditional classification algorithms.

It is obvious that the accuracy of the classification model based on the proposed method is higher than the above four traditional classification algorithms, which further proves the effectiveness of the proposed method.

## 6. Conclusions

This paper put forward a classification model based on atomic association rules, and applies it to the classification model of Tibetan medical syndrome for common plateau disease (CAG). The classification model of Tibetan medical syndrome was established through mining the strong ACAR implied between symptoms and syndrome, and the experimental results showed that the proposed method can achieve rapid classification and prediction of Tibetan medical syndrome, with fewer but more understandable rules in a shorter time, while maintaining a high accuracy, which can further provide auxiliary decision support for the diagnosis and treatment of Tibetan medical syndrome for common plateau diseases more scientifically. In the future work, we will use neural network and deep learning methods to further improve the classification accuracy.

**Author Contributions:** X.Z. performed the experiments and wrote the first version of the manuscript; L.Z. and Y.Z. made overall guidance; L.W. obtained the data set from hospital; S.W. preprocessed the original data; P.L. carried on data analysis and visualization of results; All authors read and approved the final manuscript.

**Funding:** This research was funded by The National Natural Science Foundation of China, grant number 61563044, 61762074 and 71702119; National Natural Science Foundation of Qinghai Province, grant number 2017-ZJ-902; Youth Foundation of Qinghai University, grant number 2018-QGY-7; Open Research Fund Program of State key Laboratory of Hydroscience and Engineering, grant number sklhse-2017-A-05.

**Acknowledgments:** The authors would like to thank the Tibetan Hospital of Qinghai Province for providing the clinical data of Chronic Atrophic Gastritis of common plateau disease.

**Conflicts of Interest:** The authors declare no conflict of interest.

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
