# Peer review of "Research on Classification of Tibetan Medical Syndrome in Chronic Atrophic Gastritis"

_applsci, doi:10.3390/app9081664_

Round 1

Reviewer 1 Report

A classification model based on association rules has been proposed by authors with the aim to classify Tibetan medical syndrome of common plateau disease called Chronic Atrophic Gastritis. 

The proposed approach is interesting but there are some points that the authors have to explain. The feature selection process should be better analyzed. In table 3 the authors should provide more details about significance of their results using AUC or F1-measure.

Furthermore, my suggestion is to analyze also more recent approaches about the examined topics. In particular, I suggest to cite more recent paper for analyzing the use of ontologies in e-health applications and analyze how the Chronic Atrophic Gastritis spreads out among patients:

1) A lexicon-grammar based methodology for ontology population for e-health applications. In 2015 Ninth International Conference on Complex, Intelligent, and Software Intensive Systems (pp. 521-526). IEEE.

2) Diffusion algorithms in multimedia social networks: a preliminary model. In Proceedings of the 2017 IEEE/ACM International Conference on Advances in Social Networks Analysis and Mining 2017 (pp. 844-851). ACM.

Finally, I suggest to perform a linguistic revision

Author Response

Dear reviewer:

I am very grateful to your comments for the manuscript. According with your advice, we amended the relevant part in the original manuscript. Here, we attached revised manuscript in the formats of both PDF and MS word for your approval. The revised manuscript with the correction sections was marked in red. A document answering every question to the comments was also summarized and enclosed. Should you have any questions, please contact us without hesitate.

Sincerely yours

Xiaolan Zhu

Department of Computer Technology and Applications

Qinghai University

Reviewer 2 Report

well written paper, needs some minor English correction

Author Response

(The authors gave the same response as above.)
